# The Theory of *Carcino-Evo-Devo* and Its Non-Trivial Predictions

**DOI:** 10.3390/genes13122347

**Published:** 2022-12-12

**Authors:** A. P. Kozlov

**Affiliations:** 1Vavilov Institute of General Genetics, Russian Academy of Sciences, 3 Gubkina Street, 117971 Moscow, Russia; contact@biomed.spb.ru; 2Peter the Great St. Petersburg Polytechnic University, 29 Polytechnicheskaya Street, 195251 St. Petersburg, Russia; 3Biomedical Center, 8 Viborgskaya Street, 194044 St. Petersburg, Russia

**Keywords:** *evo-devo*, *carcino-evo-devo*, tumor neofunctionalization, non-trivial predictions

## Abstract

To explain the sources of additional cell masses in the evolution of multicellular organisms, the theory of *carcino-evo-devo*, or evolution by tumor neofunctionalization, has been developed. The important demand for a new theory in experimental science is the capability to formulate non-trivial predictions which can be experimentally confirmed. Several non-trivial predictions were formulated using *carcino-evo-devo* theory, four of which are discussed in the present paper: (1) The number of cellular oncogenes should correspond to the number of cell types in the organism. The evolution of oncogenes, tumor suppressor and differentiation gene classes should proceed concurrently. (2) Evolutionarily new and evolving genes should be specifically expressed in tumors (*TSEEN* genes). (3) Human orthologs of fish *TSEEN* genes should acquire progressive functions connected with new cell types, tissues and organs. (4) Selection of tumors for new functions in the organism is possible. Evolutionarily novel organs should recapitulate tumor features in their development. As shown in this paper, these predictions have been confirmed by the laboratory of the author. Thus, we have shown that *carcino-evo-devo* theory has predictive power, fulfilling a fundamental requirement for a new theory.

## 1. Introduction

Additional cellular masses with high biosynthetic and morphogenetic potential are necessary for the evolution of multicellular organisms, especially in the line Deuterostomia—Chordata—Vertebrata. The origin of such additional cellular masses is unclear [1].

In multicellular organisms, cell division is regulated by functional feedbacks. Formation of additional cell masses means escape from the regulatory control. Unregulated cell division is one of the features of a tumor.

The hypothesis of the evolutionary role of hereditary tumors as the source of additional cell masses in evolution was first formulated in the author’s 1979 paper [2]. Since then, the concept of the positive evolutionary role of tumors has been developed in a series of publications [3,4,5,6,7,8]. In the book *Evolution by Tumor Neofunctionalization* [9], with its twelve chapters and over one thousand references, this concept started to gain the shape of the theory, which was further developed in subsequent publications [10,11,12,13,14]. In a 2019 paper, the author called this new theory the “*carcino-evo-devo*” theory to stress the role of heritable tumors in the evolution of development [11].

The important demand for a new theory in experimental science is the capability to formulate non-trivial predictions, which can be experimentally confirmed. The formulation of predictions is largely determined by the main hypothesis of a theory. The main hypothesis of *carcino-evo-devo* theory is the hypothesis of evolution by tumor neofunctionalization.

The hypothesis of evolution by tumor neofunctionalization in [9] was defined as follows:


*“Tumors are the source of extra cell masses, which may be used in the evolution of multicellular organisms for the expression of evolutionarily new genes, for the origin of new differentiated cell types with novel functions and for building new structures, which constitute evolutionary innovations and morphological novelties”.*



*Hereditary tumors may play an evolutionary role by providing conditions (space and resources) for the expression of genes newly evolving in the DNA of germ cells. As a result of expression of novel genes, tumor cells may acquire new functions and differentiate in new directions, which may lead to the origin of new cell types, tissues and organs. New cell type is inherited in progeny generations due to genetic and epigenetic mechanisms similar to those for pre-existing cell types.*


Tumors at the early stages of progression, benign tumors, pseudoneoplasms and tumor-like processes, which provide evolving multicellular organisms with extra cell masses functionally unnecessary to the organism, are considered as potentially evolutionarily meaningful. Malignant tumors at the late stages of progression, however, are not” [9].

The main hypothesis of *carcino-evo-devo* theory helped to formulate several non-trivial predictions.

## 2. Non-Trivial Predictions of the *Carcino-Evo-Devo* Theory

Several non-trivial predictions were formulated, four of which will be discussed in the present paper:(1)The number of cellular oncogenes should correspond to the number of cell types in the organism. Evolution of oncogenes, tumor suppressor and differentiation gene classes should proceed concurrently.(2)Evolutionarily new and evolving genes should be specifically expressed in tumors (*TSEEN* genes).(2’)The whole classes of genes with tumor-specific expression may be evolutionarily novel:
CT antigen genes;HERVs;ncRNA genes;pan-cancer genes.
(3)Human orthologs of fish *TSEEN* genes should acquire progressive functions connected with new cell types, tissues and organs.(4)Selection of tumors for new functions in the organism is possible. Evolutionarily novel organs should recapitulate tumor features in their development.

## 3. Confirmation of Non-Trivial Predictions

### 3.1. The Number of Cellular Oncogenes Should Correspond to the Number of Cell Types in the Organism

Originally, this prediction was formulated as follows:

*“The evolutionary role of cellular oncogenes, or proto-oncogenes might consist in sustaining a definite genetically determined level of autonomous proliferative processes in evolving populations of multicellular organisms and in promoting the expression of evolutionarily new genes in anaplastic cells of extra cell masses. After the origin of a new cell type, the corresponding oncogene should have turned into a cell type-specific regulator of cell division. If such scenario is true, then the number of different proto-oncogenes should be about 200—in accordance with the number of cell types in the multicellular organism”*.[4]

When this prediction was first formulated, only about a dozen oncogenes and two hundred cell types were known.

In 1996, the author wrote on the same prediction:

*“The evolutionary role of cellular oncogenes may consist in sustaining the definite level of autonomous proliferative processes in the evolving populations of organisms and in promoting the expression of evolutionarily new genes. After the origin of a new cell type, the corresponding oncogene should have turned into a cell type-specific regulator of cell division and gene expression. If true, the number of cellular oncogenes should correspond to the number of cell types in higher animals (10-fold higher than the 20 or so limit predicted a few years ago and now already about 70)”*.[6]

Determination of the correct number of cellular oncogenes took thirty years of work by thousands of scientists in hundreds of labs, and the work is being continued. In 2019, we performed a comparative analysis of the discovered numbers of oncogenes and cell types [15].

The published estimates of the number of cell types in humans produced numbers ranging from 240 [16,17,18] to 411 [19] cell types.

At the same time, the TAG database contained 246 oncogenes and the COSMIC database had 312 cancer genes, which approximately corresponded to the number of cell types. The number of oncogenes in the other multicellular organisms for which such information was available also corresponded to the number of cell types in these organisms [15]. Thus, the prediction about correspondence of the number of cellular oncogenes to the number of cell types is confirmed.

#### Evolution of Oncogenes, Tumor Suppressor and Differentiation Gene Classes Should Proceed Concurrently

The author further hypothesized that at least three different classes of genes are necessary for the origin and evolutionary enhancement of functional molecular feedback loops in a new cell type during evolution—oncogenes, tumor suppressor genes (TSG) and evolutionarily novel genes, which determine new functions (Figure 1).

Such a relationship predicted concurrent evolution of these gene classes. In order to prove this prediction, we performed a special study of phylogenetic distribution of orthologs of human oncogenes, tumor suppressor genes and differentiation genes. We have shown that gene age distribution curves of oncogenes, tumor suppressor genes and differentiation genes overlap. They form a cluster with perfect (100%) bootstrap reliability which confirms the coevolution of these gene classes. Moreover, we found intersections between these classes, i.e., some genes which belong to several classes (onco × diff, TSG × diff). We found that TGFβ gene belongs to all three classes, i.e., it is the triple function gene. This suggests functional co-evolution on a single gene level [15]. Thus, the prediction about concurrent evolution of oncogenes, tumor suppressor and differentiation gene classes is confirmed.

### 3.2. Evolutionarily New and Evolving Genes Should Be Specifically Expressed in Tumors (TSEEN Genes)

The important non-trivial prediction of the main hypothesis that evolutionarily new genes should be specifically expressed in tumors is contained in the formulation of the hypothesis. This prediction has been addressed in many of our papers [10,15,20,21,22,23,24,25,26,27,28,29,30,31,32,33,34,35,36,37].

Before the era of genomics came, it was not possible to prove this prediction because the sensitivity of methods was low (see e.g., [20]).

In our genomic studies that followed, the age of the gene was defined by the most recent common ancestor on the human evolutionary timeline which contained genes with similar sequences, i.e., with a significant BLAST score (or HMMER E-value) [38].

The age of the gene class was described by distribution of ages of genes belonging to this gene class. For convenience, the age of the gene class was measured numerically in million years (Ma) at the median of distribution, i.e., at the time point on the human evolutionary timeline which corresponds to the origin of 50% of orthologs of the functional gene class [15].

Tumor specificity of gene expression was studied using subtractions of all known normal cDNA libraries from all tumor DNA libraries [21,24]. We called this approach “the global subtraction”. It stemmed from our earlier saturation hybridization experiments in which we used combined RNA preparations from all experimentally available tissues of rat [20]. With the development of in silico databases, for global subtractions we used databases such as GTEx and TCGA. Genes obtained as a result of such subtractions may be called the “pan-cancer genes”.

Tumor specificity of gene expression was experimentally confirmed using PCR on normal and tumor cDNA panels (reviewed in [9,10]).

We described several evolutionarily new and young human genes with tumor-specific or tumor-predominant expression, and the whole classes of such genes. The author called them *TSEEN* genes—tumor specifically expressed, evolutionarily novel genes (reviewed in [9,10]).

#### 3.2.1. Single *TSEEN* Genes

*PBOV1* gene. In our 2013 paper [31], we described the de novo origin of the human *TSEEN* protein coding gene, *PBOV1*. It was among the first de novo originated human genes described in the literature (see discussion in [39]).

We noticed this gene among human genes without orthologs in the mouse and dog genomes in the paper of Clamp and co-authors [40] and we studied its evolutionary novelty more carefully. We found that *PBOV1* is a single-exon gene, and its ORF encodes a 135-aa protein; the ORF originated in humans through a series of frame-shift and stop codon mutations; 80% of the protein sequence is unique to humans; the protein existence is confirmed by Western blot and MS/MS identifications; the protein lacks any annotated or predicted domains; over 60% of the protein is predicted to be disordered; a protein-coding sequence is not conserved, Ka/Ks ~ 1.0, indicating that the amino acid sequence evolved neutrally; and Gene Ontology (GO) shows few functions. These findings strongly suggested a very recent de novo evolutionary origin for the *PBOV1* gene [31].

The *PBOV1* gene was originally discovered by An and co-authors, who have shown its overexpression in prostate, breast and bladder cancer [41]. In our paper, we have shown that the *PBOV1* gene is expressed in a much broader range of tumors [31]. High levels of PBOV1 expression are connected with a favorable clinical outcome for breast cancer and glioma [31]. We hypothesized that *PBOV1* protein acts as an immunological tumor suppressor [31].

Similar results have been obtained by other authors using an ovarian cancer model [42].

However, *PBOV1* promotes prostate cancer [43] and can be a biomarker for more advanced prostate cancer [44]. High *PBOV1* expression level is also a marker of poor prognosis for patients with hepatocellular carcinoma (HCC) and therapeutic target for HCC [45,46]. Its function in HCC may be connected with its role in epithelial-to-mesenchymal transition [45,47]. *PBOV1* rs6927706 polymorphism is linked to the development of breast cancer [48].

Therefore, depending on context, *PBOV1* can serve either as a tumor suppressor or an oncogene. Similar situations will be discussed below for other genes.

PBOV1 may also have other functions [49,50].

*ELFN1-AS1* gene. The other *TSEEN* gene—*ELFN1-AS1* was described by us as a novel primate gene expressed predominantly in tumors [32]. It has an extensive literature now as an lncRNA oncogene in many tumors, with appropriate references to our original paper as the first description of this gene [51,52,53,54]. In a recent review, it is presented as a human-specific de novo originated gene [55].

The *TSEEN* nature of *PBOV1* or *ELFN1-AS1* genes was studied with two different approaches. In the case of *PBOV1*, we first studied in silico the evolutionary novelty of the gene. After it was established, we studied experimentally the tumor specificity of its expression using cDNA panels from normal and tumor tissues, and we found the broad range of tumors in which *PBOV1* was specifically expressed [31].

*ELFN1-AS1* was found among EST sequences obtained by global subtraction of all known normal cDNA libraries from all tumor cDNA libraries (not the pairwise comparison of each normal tissue and corresponding tumor, as it is usually done) [21,24]. After the tumor specificity of the expression of the gene was confirmed experimentally [22], we studied its evolutionary age and discovered that it was novel for primates [32].

The de novo origin is one of fundamental modes of gene origin (see [9] for discussion). It is difficult to prove de novo origin, however. Several papers have double-checked the de novo genes described by other authors [39,55,56]. Two papers [39,55] studied the de novo origin of *PBOV1* and *ELFN1-AS1* genes again. Both genes have successfully passed the scrutiny. This confirms the correctness of our approach to *TSEEN* genes discovery.

In [55], though, there is serious inaccuracy in citation: the author did not correctly cite the paper in which de novo origin of the *PBOV1* gene was described. An and co-authors in their paper [41] did not describe the de novo origin of the *PBOV1* gene. Other examples of individual *TSEEN* genes/sequences described in the lab of the author are presented in [10].

#### 3.2.2. *TSEEN* Gene Classes

CT genes. We have also described the evolutionary novelty of a whole class of genes—cancer/testis antigen (CT) genes [30]. The evolutionary novelty of CT genes was later confirmed by other authors with appropriate reference to our priority [57].

The reasoning for prediction of the evolutionary novelty of CT genes was as follows. The author asked the question—“Why cancer/testis?”, why such unusual similarity of expression patterns in testis and different cancers? This question had never been appropriately addressed. The answer was hinted at by the fact that evolutionarily novel genes originate in the germ cells, and the germ cells are in the testis. Many genes originate through retroposition mechanism, which is why their expression in the testis is necessary. According to our hypothesis, evolutionarily novel genes should be expressed in tumors. Hence, CT genes should be young or novel genes, exactly what we found in [30]. The same line of reasoning explains the broader phenomenon of cancer/testis/brain genes, i.e., through the suggestion that an evolutionarily novel organ, the human brain, originated from a tumor-like structure (see more detailed discussion in the author’s book [9]).

In our recent paper, we confirmed the evolutionary novelty of CT genes using a different methodology [15].

##### Evolutionarily Novel HERVs Expressed Predominantly in Tumors

Different families of HERVs infected human ancestors and integrated their genome during different phylogenetic periods [58,59]. The author suggested that expression of HERVs in normal and tumor tissues should depend on their evolutionary novelty. The existing data on the expression of HERVs and other retrotransposons in tumors supported overexpression of many HERVs in tumors [10]. However, the data also suggested that HERV-derived RNAs were more widely expressed in normal tissues than originally anticipated. The author further suggested that the evolutionarily youngest HERVs should be expressed more specifically in tumors. There are very young HERV families, e.g., HERV-K HML-2, with estimated time of infection less than 1 Ma [60]. HERV-K HML-2 proviruses located on the X chromosome were chosen because the burst of X-linked gene originations happened after the split of human and chimp [61].

According to our prediction, HERV-K HML-2 located on X chromosome (HERV-K HML-2 (X)) should have higher expression in tumor tissues. The expression level of 12 HERV-K HML-2 (X) sequences were analyzed in normal tissues (lung, colon and lymphocytes) and in tumors (small cell carcinoma, colon cancer and acute myeloid leukemia). We found that the expression level of HERV-K HML-2 sequences located on the X-chromosome was dramatically higher in tumors as compared with normal tissues [62]. Therefore, a new family of *TSEEN* genes –HERV-K HML-2 (X)—was described as predicted.

ncRNA genes/sequences. About half of the tumor-specific sequences produced by global subtractions [21,24] lack long reading frames and may be referred to non-coding RNAs [10]. Among them, the *ELFN1-AS1* gene was discovered as a candidate microRNAgene [32], but later was referred to lncRNA genes [63,64]. We also described a new long non-coding RNA gene—*OTP-AS1*—which belongs to cancer/testis genes and is evolutionarily novel for eutherians, with almost 100% tumor specificity of expression in many tumors [22,36,65]. A class of ncRNA genes obtained by global subtractions is the youngest class of human genes [15]. Noncoding RNA genes may represent proto-genes evolving to new organismal functions. Proto-genes are defined as gene precursors, which have not acquired functions yet [66].

##### The Evolutionary Novelty of Tumor-Specifically Expressed Sequences Obtained by Global Subtraction (Pan-Cancer Genes)

In [15], the evolutionary ages of different gene classes have been compared.The protein-coding sequences have been studied by Protein Historian tool, and the non-coding sequencesby BLAST algorithm and the original Python script. The curves of phylogenetic distribution for orthologs of the different gene classes have been generated. They formed three clusters. Cluster I contained the oldest gene classes—housekeeping genes, oncogenes, differentiation genes and tumor suppressor genes. Cluster II contained homeobox genes, apoptosis genes, autosomal CT genes and protein coding genes from [21,24]. Cluster III was formed by the youngest gene classes—CT(X) and noncoding genes from [21,24]. Each studied gene class contained human-specific genes, even the class of house-keeping genes, which was a surprise. But the largest proportion of genes evolutionarily novel for humans was in globally subtracted noncoding RNA genes, located on the X chromosome (25%). Therefore, the results showed that noncoding tumor-specific sequences obtained by global subtraction form the youngest gene class in humans [15].

The global subtraction of normal and tumor cDNA libraries produced not only genes evolutionarily novel or young for humans. Using this approach, we described a new long non-coding RNA gene—*OTP-AS1*– novel to eutherians [36]. We also described the high tumor specificity of expression of human Brachyury homolog [26,67], orthologs of which originated in fishes. The product of this gene turned out to be a promising antigen for tumor vaccines. The respective US patent has been obtained [68]. About thirty clinical trials of tumor vaccines based on Brachyury are currently under way (ClinicalTrials.gov). Brachyury homolog participates in animal organs’ formation and may belong to the *carcino-evo-devo* genes discussed below.

#### 3.2.3. Human *TSEEN* Protein-Coding Genes Database

In 2021, we registered a *TSEEN* genes database [69]. This database was obtained in silico using the following databases and tools: GTEx database (RNASeq data of 53 normal tissues from 100 patients who died as a result of an accident and had no serious pathologies during their life); TCGA database (Data Release 28.0, 2 February 2021; RNASeq data of total RNA samples which were isolated from 20,000 biosamples covering 33 different tumors); GENEVESTIGATOR (curated database and expression data analysis tool, which includes RNASeq data of total RNA samples from eighttypes of stem cells).

The database contains 100 genes originated in humans, 234 genes originated in primates and 119 genes originated in mammals (Figure 2).

As shown in the lower part of Figure 2, human TSEEN genes are expressed in a broad range of tumors and in a considerable proportion of patients.

### 3.3. Human Orthologs of Fish TSEEN Genes Should Acquire Progressive Functions Connected with New Cell Types, Tissues and Organs

We studied this prediction using a transgenic fish inducible-tumor model. The transgenic inducible fish tumor after regression can be considered as an approximation to an organ evolving from a tumor. In so far as many evolutionarily novel genes have no functions, we studied new gene functions by comparing fish novel genes with their human orthologs using the Gene Ontology (GO) approach. We found that orthologs of many human genes, which are involved in the development of lung, mammary gland, placenta, ventricular septum, etc., originated in fish and were expressed in fish tumors and tumors after regression [35]. These data may be regarded as direct confirmation of the main hypothesis.

Figure 3 demonstrates the sequential steps of our analysis:

From 1502 genes expressed in fish tumors and tumors after regression, using Ensemble Compara we detected 409 genes with no lamprey orthologs, i.e., tumor and tumor after regression expressed evolutionary novel *(TT_Rgr_EEN)* fish genes. From 343 human orthologs of fish *TT_Rgr_EEN* genes, using GO, we selected 12 genes with important developmental and morphogenetic functions not encountered in fish. An additional 11 human orthologs of fish *TT_Rgr_EEN* genes with functions that do not exist in fish were obtained by OMA ortholog search algorithm and GO (Figure 3 and Table 1).

The progressive functions of human orthologs presented in Table 1 do not exist in fish and will never be discovered there.

Highly represented among fish, *TT_Rgr_EEN* genes are protein kinase, DNA binding and transcriptional functions, the most common domains encoded by cancer genes [35]. Other authors have also demonstrated oncogene properties in some evolutionarily novel genes [57]. The progressive functions of human orthologs have been added to fish proto-genes in the course of evolution and involved more organismic than molecular functions [35]. The novel progressive functions in human orthologs balanced the original oncogenic potential of fish proto-genes.

Some human orthologs of fish genes from Table 1 are involved in the development of several progressive features in humans. On the other hand, the development of some human progressive traits involves several human genes from this table [35]. The gene network that participates in mammalian adipose organ development will be described below.

The author suggests calling genes that originated from ancestral *TSEEN* genes and acquired progressive functions the “*carcino-evo-devo*” genes. This term was originally used as a synonym of *TSEEN* genes in [9]. In the present meaning, it was first used in [35] to stress their role in *evo-devo* and connection to carcinoembryonic antigen genes [70].

#### Conclusion on Predictions 2 and 3: TSEEN Genes—A New Biological Phenomenon and the Superclass of Novel and Evolving Genes Expressed in Tumors

The author has suggested considering the expression of evolutionarily novel and evolving genes in tumors as a new biological phenomenon, which is a part of a greater biological phenomenon defined by the main hypothesis, i.e., evolution by tumor neofunctionalization [9,10].

*TSEEN* genes as a new class of genes were reviewed for the first time in [9,10]. This new class of *TSEEN* genes has peculiarities when compared to other gene classes. First, it is not a functional gene class like the classes of oncogenes or differentiation genes. Evolutionarily novel genes often do not have functions yet [9]. The generic features of *TSEEN* genes are their tumor specificity of expression, and evolutionary novelty and continuing evolution towards new function acquisition.

The dual specificity of *TSEEN* genes determines two complementary ways for their study, as discussed above. Tumor specificity of expression can first be studied, followed by estimation of the evolutionary age; and vice-versa, evolutionarily novel genes can first be found, followed by determination of the tumor specificity of their expression [10,69].

Evolutionary novelty of a gene class is the relative characteristic. Even the evolutionarily oldest gene classes such as housekeeping genes contain genes originated in humans, but their proportion is extremely low [15]. Various classes of human *TSEEN* genes, for example, CT antigen genes, contain older genes, and not only human and primate specific genes. However, *TSEEN* gene classes contain the highest proportions of evolutionarily new and young genes. The median of gene age distribution for *TSEEN* gene classes demonstrates that in humans they are the youngest gene classes [15].

The main hypothesis does not specify the strict borders of the *TSEEN* gene class. As shown earlier by us and other authors, genes originate somewhat earlier than corresponding morphological structures appear [15,35,71]. Our data on fish *TSEEN* genes suggest that their human orthologs were important for the origin of progressive mammalian traits [35]. Higher prevalence of *TSEEN* genes in primates (Figure 2) suggests their role in human evolution. Human tumor-specifically expressed genes that originated in mammals have not acquired functions in normal tissues yet, which is why they were considered as evolving genes and were placed in Figure 2. Genes represented in Figure 2 demonstrate the gradient of evolutionary novelty that reflects an evolutionary continuum of evolving genes. Expression of proto-genes, young and novel genes in hereditary tumors may represent the early stages in the origin of novel gene functions. This view corresponds well to the concept of proto-gene [66] and to several models of gene origin described in [72]. *TSEEN* genes may represent a reservoir of proto-genes pervasively expressed in tumors. It would be most interesting to study the process of new progressive function acquisition by *TSEEN* genes, which have not acquired functions yet. The algorithm for such studies is suggested in [14].

Various *TSEEN* genes can acquire different functions in evolution, as shown in [35]. *TSEEN* gene classes of various phyla of organisms are different and may evolve in different directions. As shown in Figure 3, a considerable proportion of fish *TT_Rgr_EEN* genes do not have orthologs in humans.

The author suggests considering *TSEEN* genes as a new superclass of novel and evolving genes with tumor-specific expression, with several classes and families of *TSEEN* genes, which includes *TSEEN* genes of various phyla of organisms. The author has called *TSEEN* genes’ descendants, which acquired progressive developmental and morphogenetic functions during evolution, the *carcino-evo-devo* genes. The discovery of *carcino-evo-devo* genes in [35] may be considered as the direct confirmation of the main hypothesis.

The expression of *TSEEN* genes in a wide range of tumors and in a considerable proportion of patients is their important characteristic that makes them perspective targets for the development of universal tumor test systems, prophylactic and therapeutic tumor vaccines, and conventional therapeutic agents.

Other authors have recognized the connection of evolutionarily novel genes with tumors. In their reviews, they discuss the *TSEEN* genes described by us and others [39,55,72].

### 3.4. Selection of Tumors for New Functions in the Organism Is Possible. Evolutionarily Novel Organs Should Recapitulate Tumor Features in Their Development

#### 3.4.1. Selection of Tumors for New Functions

The main hypothesis predicts that hereditary tumors may be selected for new organismal functions. A special paper was devoted to confirmation of this prediction [73]. We have shown that so called “hoods” on the heads of certain varieties of goldfish are benign tumors. These tumors were selected for hundreds of years by Chinese breeders. As a result of this selection, a new organ—the hood—originated. The symmetrical shape of the hood, its benign properties and its appearance at certain stages of development are features of the normal organ. The capability for unlimited growth and histological peculiarities are tumor features. The author suggested calling such organs of dual nature the “tumor-like” organs [9].

To our knowledge, the “hood” of goldfish was the first example of artificial selection of tumors described in the literature. This was discussed in the book [9] and in a recent paper devoted to tumor-like organs [12].

Symbiovilli in voles originated by natural selection of the early stages of papillomatosis [74], and macromelanophores in swordtails by sexual selection [75]. We may conclude that the nontrivial prediction about the possibility of selection of hereditary tumors for new organismal functions works. Positive selection of many tumor-related genes in the primate lineage discussed in the book [9] supports this view.

#### 3.4.2. Evolutionarily Novel Organs That Recapitulate Tumor Features in Their Development: Mammalian Tumor-like Organs

The author addressed this topic in the book [9] and in two recently published papers devoted solely to this question [12,13]. In [12], the author was looking for tumor features of four mammalian evolutionarily novel organs: placenta, mammary gland, prostate and infantile human brain. In [13], the tumor-like features of mammalian adipose were described for the first time in the literature, a discovery that can change the medicine of obesity and cancer.

Placenta was the first organ the tumor-like features of which were recognized because it has dozens of such features (reviewed in [9,12]). The role of ancient retrovirus infection and *syncytin* gene domestication was also early recognized [76,77,78]. A special symposium was held on this particular organ and its resemblance to tumors [79]. A term “tumor-like organ” is used for placenta [9,80,81].

The mammary gland and prostate are characterized by the highest incidences of cancer. It was shown that the high incidence of cancer in these organs is connected with their evolutionary novelty [82]. Like placenta, they have a regulated invasion stage in their organogenesis. Both glands have many other similarities with tumors (reviewed in [12]) that indicate the tumor-like nature of mammary gland and prostate.

Human brain recapitulates many features of tumors and demonstrates more of its tumor-like nature during childhood and infancy. Brain tumors are the most common solid tumors and the leading cause of cancer death in children (reviewed in [9,12]).

The tumor-like features of mammalian adipose [13] will be discussed below.

One more evolutionary novel organ (or tissue) with tumor features can be added to the list of organs discussed in [12,13]: the ependymal region of the adult human spinal cord, which differs from other species and shows ependymoma-like features [83].

The features of tumor-like organs position them close to the so-called “atypical tumor organs”. The concept of atypical tumor organs is already accepted among oncologists. It reflects the complex structure of solid tumors, which consist of parenchyma with a hierarchy of cell types and stroma with connective tissue, blood vessels and accessory cells (see [84] for review). According to the main hypothesis, normal tumor-like organs could evolve from hereditary atypical tumor organs. The relationship between tumor-like organs and atypical tumor organs represents an essential part of the *carcino-evo-devo* relationship, i.e., coevolution of normal and neoplastic development [12]. The term “*carcino-evo-devo”* gave a name to the theory of the evolutionary role of hereditary tumors.

The series of our two papers devoted to mammalian tumor-like organs answered the important and frequent question: how often in evolution could tumor neofunctionalization happen? The answer is that in mammalian evolution it happened often.

#### 3.4.3. Tumor Features of Mammalian Adipose. Obesity as a Tumor-like Process

An unexpected discovery was the finding of many tumor features in mammalian adipose [13]. It was stimulated by two circumstances: the concept that several mammalian adipose tissues constitute the mammalian adipose organ, and the evolutionary novelty of this organ. Adipose is currently recognized as a metabolic and endocrine organ operating “as a structured whole” [85,86]. Although the storage of energy in lipids is evolutionarily conserved, and lipid-storing cells and proteins are ancient [87,88], the adipose organ is evolutionarily novel to mammals [89,90]. Brown adipose tissue (BAT) has not been described in fishes, amphibians, reptiles or birds, and is present only in higher mammals [91,92,93].

After understanding these things, the author started looking for tumor-like features of mammalian adipose and found many such features. Similarities of mammalian adipose with tumors include the following: the capability to unlimited expansion; lipomas as the most frequent soft tissue tumors; reversible plasticity; induction of angiogenesis; chronic inflammation; remodeling and disfunction; systemic influence on the organism; hormone production; production of miRNAs that influence other tissues; immunosuppression; DNA damage and resistance to apoptosis; infiltration of other organs and tissues; and similar drugs that may be used for treatment of obesity and cancer [13].The most luminous similarities are the capability to unlimited expansion, due to hyperplasia and hypertrophy, and infiltration of other organs and tissues. Many of the common features of tumors and the adipose organ are in the list of so-called “hallmarks of cancer” [94,95], and many of them are connected with obesity. Thus, mammalian adipose is a tumor-like organ, and obesity is a tumor-like process, a discovery that may change the landscape of public health and medicine. It means that we can probably use the approaches developed in oncology to fight obesity, and the approaches used against obesity to fight cancer.

The tumor-like nature of mammalian adipose suggests its origin from some ancestral mesenchymal tumors. The author has used our data obtained with the transgenic fish hepatoma model in [35] to explore this possibility [13]. It was found that five genes (*LEP*, *SPRY1*, *PPARG*, *ID2* and *NOTCH1*) among the 23 human orthologs of fish *TT_Rgr_EEN* genes with progressive functions not encountered in fish described in [35] also participate in mammalian adipose organ development. Progressive functions connected with the adipose organ were not mentioned in [35] because we were not interested in mammalian adipose at that time and did not look for adipose-related functions. One more gene involved in mammalian adipose development—the *CIDEA* gene—was found in [13] among 343 human orthologs of the fish *TT_Rgr_EEN* genes described in [35]. In Table 2, the functions of genes related to the mammalian adipose organ are presented:

These are progressive functions connected mainly with white adipose beiging, BAT and thermoregulation, and not encountered in fish.

Besides their function in adipose development, these genes also participate in tumor development in humans. Depending on the context, they can play tumor-promoting or tumor-suppressing roles (reviewed in [13]).

These genes also interact with each other and form a gene network. In the author’s presentation at the American Association for Cancer Research Annual Meeting 2022 (AACR 2022) [37], the following diagram describing interactions of *LEP*, *SPRY1*, *PPARG*, *ID2*, *CIDEA* and *NOTCH1* and their possible roles in tumorigenesis was presented (Figure 4):

In this gene network, *LEP* and *PPARG* play particular roles. *LEP*, which encodes leptin, became the central regulator of energy metabolism in mammals. *PPARG* is a major regulator of adipocyte differentiation and function (reviewed in [13]).

After the conference, one more member of this gene network was found among the 343 human orthologs of fish *TT_Rgr_EEN* genes—the *ZAG* or *AZGP1* gene. *ZAG* plays a key role in lipid metabolism in adipocytes and participates in white adipose tissue browning. It affects glucose metabolism and is linked to insulin resistance, and it has a role in reducing obesity and improving insulin sensitivity. *ZAG* inhibits *LEP*. *ZAG* can both stimulate and inhibit expression of *PPARG*. On the other hand, the increased expression of ZAG is influenced by *PPARG* [96,97,98,99].

Like other members of the adipose gene network, the *ZAG* gene is also involved in the development of many different types of tumors [100]. It functions mostly as a tumor suppressor gene [100,101,102,103,104], but sometimes also as an oncogene [105]. *ZAG* has a pro-carcinogenic effect on breast cancer cells and an anti-carcinogenic effect on nonmalignant cells [106].

The upgraded adipose gene network, which includes the *ZAG* gene and its interactions, is presented at Figure 5:

In the upgraded adipose gene network, the *ZAG* gene as the major lipid mobilizing factor may play the important role comparable to that of *LEP* and *PPRG*.

The work on the adipose gene network will be continued and will certainly bring to light new gene members and new interactions. However, important conclusions can be made right now.

We can summarize that orthologs of *LEP*, *NOTCH1*, *SPRY1*, *PPARG*, *ID2*, *CIDEA* and *ZAG* genes originated in fishes and were expressed in fish tumors and tumors after regression. In mammals, these genes acquired progressive functions connected with the adipose organ, form a gene network with mutual influences, and participate in tumor processes. These data suggest the tumor-like nature of mammalian adipose and support the possibility of its origin from ancestral hereditary tumors.

The most important circumstance is that all genes in the above diagram play the dual role in tumor development. Depending on the context, they play roles of either oncogenes or tumor suppressor genes. Similar duality was described for other tumor-related genes. *TGFβ* acts as tumor suppressor in normal cells and as oncogene in malignant cells [107,108]. Mouse models of *BRCA1* and *BRCA2* deficiency demonstrated a similar paradox: *BRCA*-deficient tumor cells proliferate rapidly, but developing *BRCA*-deficient embryos suffer from proliferation defect [109]. High and low levels of TNF have opposing effects on tumor growth [110]. Tumor suppressor p53 is characterized by antagonistic bifunctionality, producing both positive and negative signals on cell migration, metabolism, differentiation and survival [111]. In a recent publication, a paradoxical promotion of liver carcinogenesis by constitutive activation of p53 was demonstrated [112].

All seven members of the adipose gene network have similar antagonistic tumor duality. This may be connected with several circumstances: (1) the ancestral fish *TSEEN* genes duality; (2) the coevolution of oncogenes, tumor suppressor genes and differentiation genes; and (3) the addition in the course of evolution of domains with progressive functions to fish cancer genes. Indeed, in our previous article, we demonstrated that many genes belong to two functional classes, and *TGFβ* even has triple specificity [15]. The new progressive functions could neutralize the original tumorigenic potential of fish proto-genes and create a basis for governing homeostasis.

The phenomenon of dual functionality of tumor-related genes may have fundamental importance. The key to control over cancer and obesity may consist in understanding the antagonistic bifunctionality of tumor- and obesity-related genes and developing technologies to regulate the networks of genes with multiple antagonistic functionalities.

## 4. Conclusions: Towards a Comprehensive Theory of Evolutionary Oncology

Throughout the work on development of *carcino-evo-devo* theory, a considerable amount of effort by the author’s lab has been devoted to confirmation of non-trivial predictions, which were formulated with the help of the main hypothesis. Several predictions have been confirmed, as shown in this paper. Thus, we have shown that *carcino-evo-devo* theory has predictive power in several related areas of biology, fulfilling a fundamental requirement for a new theory.

Other monographs devoted to problems of evolutionary oncology [113,114,115] have explored other directions and do not contradict our theory.

The field of evolutionary oncology is actively developing. The author hopes that at some point we will have a comprehensive theory of evolutionary oncology.

## Figures and Tables

**Figure 1 genes-13-02347-f001:**
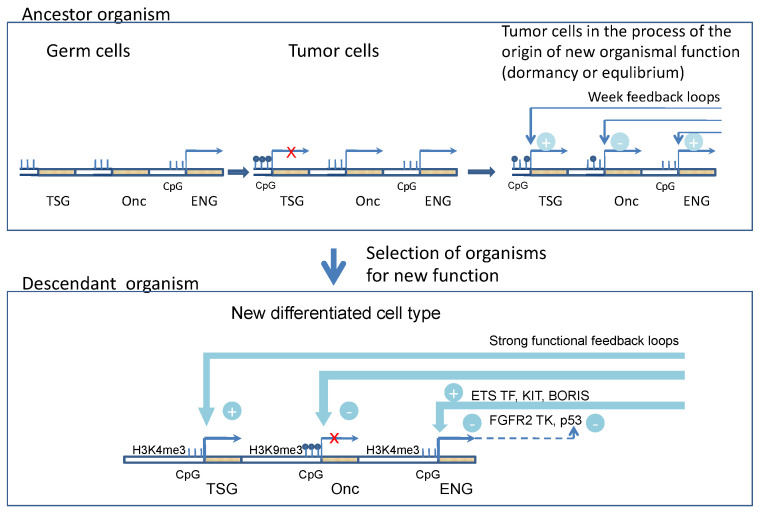
Three different classes of genes are necessary for the origin and evolutionary enhancement of functional molecular feedback loops in a new cell type during evolution—oncogenes (Onc), tumor suppressor genes (TSG) and evolutionarily novel genes, which determine new functions (ENG). Reprinted with permission from Ref. [9].Copyright 2014 Elsevier Inc.

**Figure 2 genes-13-02347-f002:**
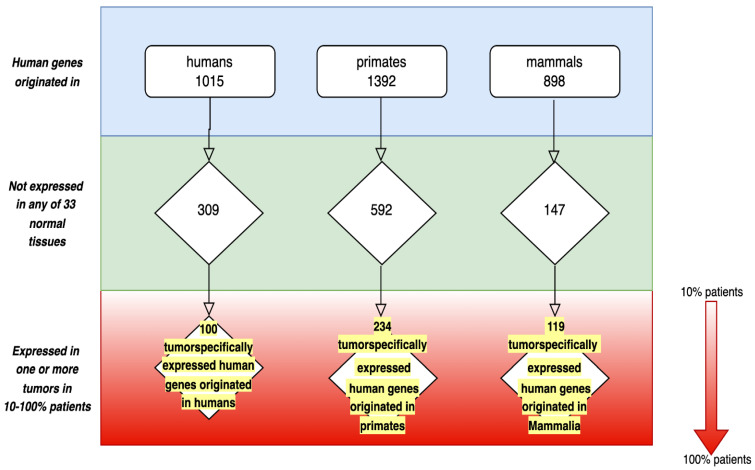
Human *TSEEN* protein-coding genes database.

**Figure 3 genes-13-02347-f003:**
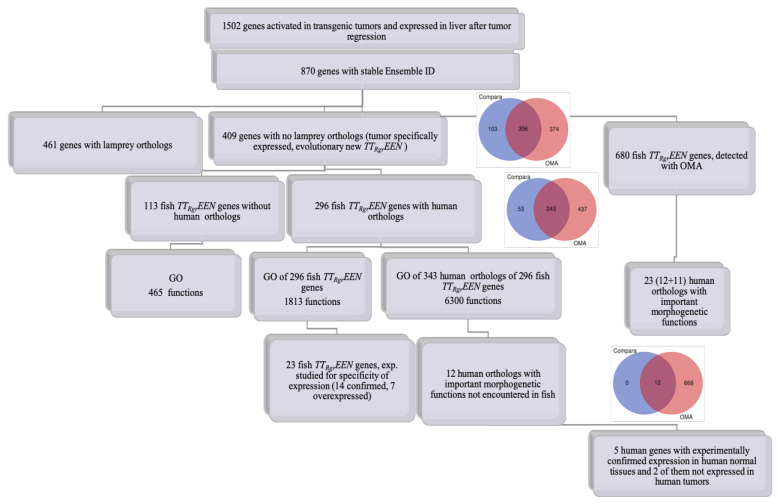
Flow diagram for the study of selected groups of zebrafish genes and their human orthologs. Reprinted with permission from Ref. [35]. Copyright 2019 of the authors.

**Figure 4 genes-13-02347-f004:**
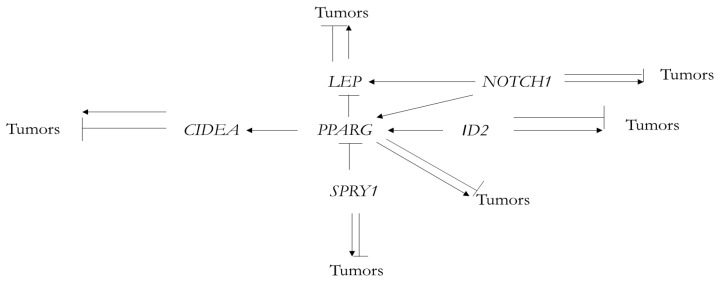
Mammalian adipose gene network originated from fish *TT_Rgr_EEN* genes, which participates in adipose organ development and tumor formation.

**Figure 5 genes-13-02347-f005:**
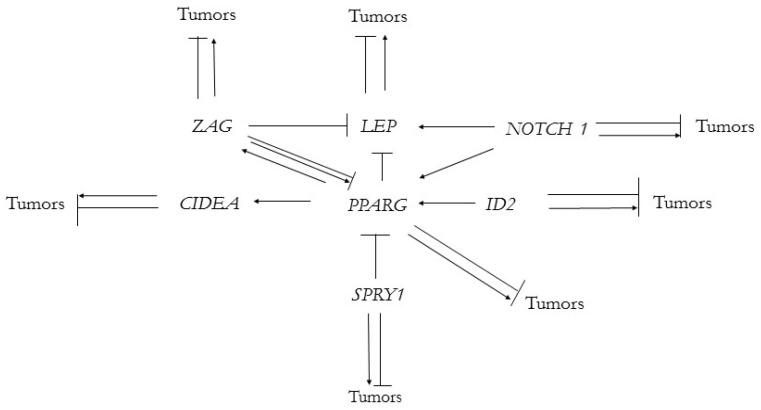
Upgraded version of adipose gene network, which includes ZAG gene.

**Table 1 genes-13-02347-t001:** Selected human orthologs of fish *TT_Rgr_EEN* genes with functions that do not exist in fish. Adapted with permission from Ref. [35]. Copyright 2019 of the authors.

Name of Gene (Fish Gene/Human Gene)	GO Domain			Selected GO Progressive Functions Not Encountered in Fish ([Fish Gene]/[Human Gene])
	Molecular Function (Fish Gene/Human Gene)	Cellular Component (Fish Gene/Human Gene)	Biological Process (Fish Gene/Human Gene)	
Fish tgfbr2b/ Human *TGFBR2*	10/18	4/12	9/84	[NO]/[bronchus development, bronchus morphogenesis, embryo implantation, in utero embryonic development, lung development, lung lobe morphogenesis, lung morphogenesis, mammary gland morphogenesis, ventricular septum development]
Fish lepa/ Human *LEP*	2/4	3/3	15/106	[NO]/[placenta development
Fish sema7a/ Human *SEMA7A*	1/3	0/4	1/16	[NO]/[olfactory lobe development
Fish klf1/Human *KLF1*	3/7	1/2	3/6	[NO]/[maternal process involved in female pregnancy
Fish ephb3a/ Human *EPHB3*	7/9	3/8	5/25	[NO]/corpus callosum development
Fish dazap1/Human *DAZAP1*	2/6	0/6	0/6	[NO]/maternal placenta development
Fish spry1/ Human *SPRY1*	0/1	1/6	6/16	[NO]/bud elongation involved in lung branching
Fish lmx1bb/ Human *LMX1B*	3/7	1/1	13/9	[NO]/in utero embryonic development
Fish nr2e1/ Human *NR2E1*	5/9	1/2	2/41	[NO]/cerebral cortex development, cerebral cortex neuron differentiation, dentate gyrus development, layer formation in cerebral cortex
Fish sobpa/ Human *SOBP*	0/2	0/1	0/5	[NO]/cochlea development
Fish ccdc40/Human *CCDC40*	0/0	4/5	11/14	[NO]/lung development
Fish fosl1a/ Human FOSL1	0/7	0/6	0/29	[NO]/placenta blood vessel development
Fish atxn1l/Human ATXN1L	1/3	1/5	0/10	[NO]/lung alveolus development
Fish id2a/Human ID2	1/3	2/4	8/56	[NO]/epithelial cell differentiation involved in mammary gland alveolus development, mammary gland epithelial cell proliferation, mammary gland alveolus development, ventricular septum development
Fish ccr11.1/Human CX3CR1	3/4	2/7	4/17	[NO]/cerebral cortex cell migration
Fish cntnap2a/Human CNTNAP2	0/2	2/15	1/8	[NO]/cerebral cortex development
Fish mycn/Human MYCN	2/7	1/3	1/20	[NO]/lung development
Fish neflb/Human NEFL	1/10	2/10	1/29	[NO]/cerebral cortex development
Fish notch1b/Human NOTCH1	3/15	1/20	15/162	[NO]/lung development
Fish reck/Human RECK	0/5	0/4	7/8	[NO]/embryo implantation
Fish srd5a1/Human SRD5A1	2/7	2/11	4/40	[NO]/cerebral cortex development
Fish wnt7bb/Human WNT7B	2/3	3/9	4/42	[NO]/trachea cartilage morphogenesis, lobar bronchus development, lung epithelium development, lung development, lung morphogenesis, chorio-allantoic fusion, embryonic placenta morphogenesis, mammary gland epithelium development
Fish pparg/Human PPARG	7/30	2/8	6/81	[NO]/placenta development

**Table 2 genes-13-02347-t002:** Functions of human orthologs of fish *TT_Rgr_EEN* genes involved in human adipose development [13].

Name of Gene	Progressive Functions Connected with Beiging, BAT and Thermoregulation, Not Encountered in Fish
*LEP*	Regulation of energy metabolism in mammals Regulation of beige/brown fat cell differentiation Lipostatic function and thermoregulation
*NOTCH1*	Regulation of adipose browning, energy metabolism and thermogenesis
*SPRY1*	Initiation and regulation of adipogenesis Maintaining proliferation and differentiation of adipose stem cells (ASCs)
*PPARG*	Differentiation of adipocytes Activation of thermogenic gene expression in brown adipocytes The role in lipodystrophy, obesity and diabetes
*ID2*	Stimulation of adipocyte differentiation and adipogenesis The role in obesity
*CIDEA*	Association with lipid droplets Regulation of lipid metabolism Regulation of adipocyte beiging

## Data Availability

Not applicable.

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
