# Peer review of "The Theory of Carcino-Evo-Devo and Its Non-Trivial Predictions"

_genes, 2022, doi:10.3390/genes13122347_

Round 1
Reviewer 1 Report
This is a nice review paper that thoroughly covered the theory of carcino-evo-devo. This is an interesting theory, which hypothesizes that new cell types originate through tumor evolution. The author has a significant contribution to the development of this theory, and this review paper provides an excellent summary of supporting evidence for this theory. Specifically, the focus was the relationship between the number of oncogenes and cell types, age of genes, and gene expression patterns. This review paper will be a good reference for the carcino-evo-devo theory.
1. One of the characteristics of tumors is a large number of mutations. This means that tumor cells have different genetic profiles from the other normal cells, and it does not inherit in general. I guess the carcino-evo-devo theory is valid under a particular situation. Some comments will be useful.
2. The content of Table 1 is a figure. It should be treated as a figure.
Author Response
- Yes, carcino-evo-devo theory is valid for hereditary tumors at early stages of progression, but not for malignant tumors at late stages of progression, which kill their hosts. The author added one more paragraph of the definition of the main hypothesis from part 10.1 of the book [Kozlov, 2014] in response to this comment.
- Table 1 is designated as Fig.2.
Reviewer 2 Report
It is well known that tumor is a hereditary disease. Hereditary tumours may have played an important role in the early stages of metazoan evolution, providing additional cell clumps for the origin of new cell types, tissues and organs. Previous research by Kozlov and his colleagues found that the evolution of oncogenes, tumor suppressor genes, and differentiation genes occurred simultaneously, and identified a new class of genes (TSEEN) specifically expressed and evolved in tumors. Their work has important social implications as these TSEEN genes could be used to build new cancer detection systems and anti-tumor vaccines.
In this review, the authors discuss four of these predictions based on the tumor evolution-differentiation theory. It is proved that the theory of carcinogenesis transformation has the ability to predict and meet the basic requirements of the new theory.
Minor comments
1.What does the author think about the role of epigenetic inheritance in tumor evolution?
2.Headings 1) and 2) are missing. The author should add some first and second level headings so that the reader can read more clearly.
Author Response
1. In the formulation of the main hypothesis, it is written that "the new cell type is inherited in progeny generations due to genetic and epigenetic mechanisms similar to those for pre-existing cell types." Fig.1 demonstrates the evolution of molecular feedback loops, which are necessary to regulate new gene expression. In the book [Kozlov, 2014] epigenetically caused tumor-related abnormalities and meiotic transmission of epigenetic signals are discussed in detail in parts 10.4 and 10.5.
2. Headings 1) and 2) are added, first and second level headings are also added.
Reviewer 3 Report
I found the review informative and generally well written. I have just a few points for the author. The first sentence of the abstract is exactly the same with the first sentence of the intro. Please change. Another thing that I didn't like is the use of the phrase"my laboratory", "my paper" etc. Please change appropriately.
Author Response
- The first sentence of the abstract is deleted.
- "My laboratory", "my paper" etc. are changed as "the author's laboratory", "the author's paper", etc. The author completely agrees with these changes, which improved the quality of the manuscript.